# D-LL-31 enhances biofilm-eradicating effect of currently used antibiotics for chronic rhinosinusitis and its immunomodulatory activity on human lung epithelial cells

Saharut Wongkaewkhiaw[1], Suwimol Taweechaisupapong[2], Sanguansak Thanaviratananich[3], Jan G. M. Bolscher[4], Kamran Nazmi[4], Chitchanok Anutrakunchai[2], Sorujsiri Chareonsudjai[1,2], Sakawrat Kanthawong[1,2]*

1 Department of Microbiology, Faculty of Medicine, Khon Kaen University, Khon Kaen, Thailand, 2 Biofilm Research Group, Faculty of Dentistry, Khon Kaen University, Khon Kaen, Thailand, 3 Department of Otorhinolaryngology, Faculty of Medicine, Khon Kaen University, Khon Kaen, Thailand, 4 Department of Oral Biochemistry, Academic Centre for Dentistry Amsterdam (ACTA), University of Amsterdam and Vrije Universiteit Amsterdam, Amsterdam, The Netherlands

* sakawrat@kku.ac.th

## Abstract

Chronic rhinosinusitis (CRS) is a chronic disease that involves long-term inflammation of the nasal cavity and paranasal sinuses. Bacterial biofilms present on the sinus mucosa of certain patients reportedly exhibit resistance against traditional antibiotics, as evidenced by relapse, resulting in severe disease. The aim of this study was to determine the killing activity of human cathelicidin antimicrobial peptides (LL-37, LL-31) and their D-enantiomers (D-LL-37, D-LL-31), alone and in combination with conventional antibiotics (amoxicillin; AMX and tobramycin; TOB), against bacteria grown as biofilm, and to investigate the biological activities of the peptides on human lung epithelial cells. D-LL-31 was the most effective peptide against bacteria under biofilm-stimulating conditions based on $IC_{50}$ values. The synergistic effect of D-LL-31 with AMX and TOB decreased the $IC_{50}$ values of antibiotics by 16-fold and could eliminate the biofilm matrix in all tested bacterial strains. D-LL-31 did not cause cytotoxic effects in A549 cells at 25 µM after 24 h of incubation. Moreover, a cytokine array indicated that there was no significant induction of the cytokines involving in immuno-pathogenesis of CRS in the presence of D-LL-31. However, a tissue-remodeling-associated protein was observed that may prevent the progression of nasal polyposis in CRS patients. Therefore, a combination of D-LL-31 with AMX or TOB may improve the efficacy of currently used antibiotics to kill biofilm-embedded bacteria and eliminate the biofilm matrix. This combination might be clinically applicable for treatment of patients with biofilm-associated CRS.

## Introduction

Chronic rhinosinusitis (CRS) is defined as chronic infection and inflammation of the upper airways involving the nasal cavity and paranasal sinuses with a duration of at least 12

**Data Availability Statement:** All relevant data are within the manuscript and its Supporting Information files.

**Funding:** SK was supported by grant fund under Khon Kaen University, Thailand (61003402 and 6200021004) https://www.kku.ac.th/ SW was supported by grant fund under Faculty of Medicine, Khon Kaen University, Khon Kaen, Thailand https://www.md.kku.ac.th/ The funders had no role in study design, data collection and analysis, decision to publish, or preparation of the manuscript.

**Competing interests:** The authors have declared that no competing interests exist.

consecutive weeks [1]. This disease significantly impacts the quality of life of patients and imposes a socio-economic burden on the community [2]. CRS is now documented as an extremely widespread inflammatory disease with an estimated prevalence of 12% across the world [3]. CRS has been subdivided into 2 major categories: CRSwNP (chronic rhinosinusitis with nasal polyps present) and CRSsNP (chronic rhinosinusitis without nasal polyps) [1, 3].

The U.S. National Institutes of Health estimated that at least 65% of all microbial infections are associated with biofilms [4]. Recently, many bacterial biofilm-associated CRS cases have been reported [5, 6]. Of these, 80% were found to have micrographic indication of biofilms on the nasal mucosa after undergoing sinus surgery [7]. Consequently, the management of biofilm-associated infections is problematic as they are difficult to prevent, diagnose, and treat [8].

A dramatically reduced susceptibility to conventional antibiotics (up to 1000 times) and to host immune responses is typical of biofilm-associated infections [8, 9]. Biofilm-embedded bacteria in CRS patients exhibit resistance to various groups of antibiotics such as tobramycin (TOB), moxifloxacin and also beta-lactam antibiotics; cephalosporins, ampicillin and amoxicillin (AMX) [10, 11]. Biofilm is increasingly implicated in relapse, persistence and severity of certain CRS cases and in recalcitrant infections [12, 13]. To treat biofilm-based antibiotic resistance, there is an increasing need to design novel therapeutic agents and/or discover alternative agents that improve the bactericidal activity of the currently used antibiotics.

Antimicrobial peptides (AMPs) are host-defense peptides with a broad spectrum of biological activities against a wide range of microorganisms (e.g. bacteria, fungi and viruses) [14–16], which are regarded as promising additions to antibiofilm strategies. Human cathelicidin-derived peptide LL-37 is a 37-amino-acid cationic peptide generated by extracellular cleavage of the C-terminal end of the hCAP18 protein (18 kDa) by serine proteases [17]. This peptide exhibits strong antibiofilm capacity against various pathogens such as *Candida albicans*, *Staphylococcus aureus*, and *Escherichia coli* [18]. Also, a truncated form of LL-37, produced by organic chemical synthesis without the six C-terminus residues (LL-31), has shown efficacy against biofilm-forming *Burkholderia pseudomallei* [19]. The substitution of L- by D-amino acids in these peptides improves their stability, enhances their antimicrobial activity and reduces toxicity [20]. The latest research has also found that D-LL-31 exhibits high potency against *B. pseudomallei* biofilm in static and flow-cell systems without causing toxicity to human red blood cells (hRBCs) [15].

A patient with CRS generally exhibits an inflammatory immune response [1, 3]. Several studies have shown that elevated levels of pro-inflammatory cytokines, interleukin-4 (IL-4), IL-6 and interferon-gamma (IFN-γ) result in decreased specialized tight junctions, which could contribute to the defective function of a physical barrier in the mucosal tissue [21, 22]. In addition, recruitment of B cells by tumor necrosis factor alpha (TNF-α) has also been thought to play a role in the pathogenesis of CRS [23]. The pro-inflammatory milieu might function to impair the mucosal barrier and promote chronic inflammation by allowing microbes, antigens, and allergens across the mucosal tissue, where they can trigger or promote immune responses [21, 22].

The chronic inflammation is typical of CRS [1, 3]. An imbalance between the matrix metalloproteinases (MMPs) and their tissue inhibitors (TIMPs) was found to be involved in pathological tissue remodeling in CRS patients [24]. The elevated expression of MMP-2, -7 and -9 [24–26] and significant decrease of TIMP-1 and -2 [25, 27] have been frequently found in CRS patients. This imbalance results in excessive degradation of airway extracellular matrix (ECM), pseudocyst formation, albumin deposition and edema in patients with CRS [28]. Thus, the immunomodulatory effect of any new therapeutic agent should be investigated before clinical use.

In this study, the bactericidal effects of LL-37, LL-31 and their D-enantiomeric forms (D-LL-37 and D-LL31) against bacteria isolated from CRS patients grown as biofilm were investigated alone and in combination with conventional antibiotics. We found that D-LL-31 has a strong killing effect against all tested bacteria under biofilm-stimulating conditions. It also enhanced antimicrobial activity of the commonly used antibiotics to combat biofilm-embedded cells and to eradicate the biofilm matrix. Additionally, D-LL-31 does not cause the induction of the cytokines involved in immunopathogenesis of CRS on human lung epithelial cells.

## Materials and methods

### Peptide synthesis

The human cathelicidin peptides LL-37, LL-31 and D-enantiomers (D-LL-37 and D-LL-31) (Table 1) were synthesized by solid-phase peptide synthesis using fluoren-9-ylmethoxycarbo-nyl (Fmoc) chemistry with a Siro II synthesizer (Biotage, Uppsala Sweden) according to the manufacturer's protocol. Labeling of D-LL-31 with 5,6-carboxytetramethylrhodamine (TAMRA) was carried out in-synthesis using an additional Fmoc-Ahx-OH (NovaBiochem) at the N-terminus. Peptides were purified to at least 95% purity by preparative reversed-phase HPLC on a Dionex Ultimate 3000 system (Thermo Scientific, Breda, the Netherlands). The authenticity was confirmed by mass spectrometry with a Microflex LRF MALDI-TOF (Bruker Daltonik GmbH, Bremen, Germany) essentially as described previously [29, 30]. Molar concentrations were calculated based on their weight.

### Ethics statement

All patients provided signed consent in accordance with the ethical principles relating to bio-medical research involving human subjects as adopted by the 18[th] World Medical Association General Assembly, Helsinki (1964) and the ICH Good Clinical Practice Guidelines. This study was approved by the Center for Ethics in Human Research, Khon Kaen University, reference number: HE561075.

### Sample collection and bacterial identification

Washings from sinus irrigation and sinus mucosa samples were collected from patients with chronic rhinosinusitis, defined according to the criteria of the 2003 Chronic Rhinosinusitis Task Force [31]. These patients were potential candidates for endoscopic sinus surgery in the otolaryngology clinic, Srinagarind Hospital, Khon Kaen University, Thailand.

The sinus lavage was cultured for aerobic and facultative anaerobic bacteria in Luria-Ber-tani broth (Himedia®, Mumbai, India) and Thioglycollate broth (Himedia®). Then, the bacteria were grown on 5% sheep-blood agar, chocolate agar or MacConkey agar (MAC)

**Table 1. Sequences and characteristics of the peptides investigated.**

| Peptides[a] | Sequences | Mol. wt. | Charge[b] |
|---|---|---|---|
| LL-37 | LLGDFFRKSKEKIGKEFKRIVQRIKDFLRNLVPRTES | 4494 | 6+ |
| D-LL-37 | *LLGDFFRKSKEKIGKEFKRIVQRIKDFLRNLVPRTES* | 4494 | 6+ |
| LL-31 | LLGDFFRKSKEKIGKEFKRIVQRIKDFLRNL | 3824 | 6+ |
| D-LL-31 | *LLGDFFRKSKEKIGKEFKRIVQRIKDFLRNL* | 3824 | 6+ |

[a]The purity of peptides was at least 95% and the authenticity of the peptides was confirmed by MALDI-TOF mass spectrometry.

[b]Calculated net charge at neutral pH.

(Himedia®). Cultures on MAC and 5% sheep-blood agar were incubated at 37°C for 24 h, whereas chocolate agar was kept under 5% $CO_2$ at 37°C and results observed after 24, 48 and 72 h. Afterward, a single colony of each sample was identified to genus and species according to standard bacteriological methods.

### Biofilm observation on sinus mucosal tissues of CRS patients

Biofilm architecture on sinus mucosal tissues of CRS patients was observed using scanning electron microscopy (SEM). The tissue samples were taken from diseased maxillary or ethmoid sinuses during functional endoscopic sinus surgery. Tissues were fixed by 4% paraformaldehyde/1.25% glutaraldehyde in PBS with 4% sucrose, pH 7.2, and stored at 4°C. After three PBS washes, samples were post-fixed with 2% osmium tetroxide ($OsO_4$) on a rotator for 1 h. The dehydration of the tissue samples was subsequently performed using a graded series of 70% to 100% ethanol concentrations. Samples were critical-point dried in $CO_2$, mounted on scanning electron microscopy stubs, sputter coated with gold palladium (K500X sputter coater, EMITECH, Quorum technologies LTD, UK), and examined using a field emission gun scanning electron microscope (S-3000N, HITACHI, Japan).

### Bacterial strains and growth conditions

Three clinical bacterial isolates, *Klebsiella pneumoniae*, *Pseudomonas aeruginosa* and *Staphylococcus epidermidis*, were selected from patients no. 1, 3 and 5, respectively (Table 2) according

**Table 2. Characteristics of the 20 patients with CRS.**

| No. | Age[a]/gender[b] | Microorganisms isolated from maxillary and ethmoid sinuses[c] | Total No. of isolate | Biofilm[d] |
|---|---|---|---|---|
| 1 | 61/f | *P. aeruginosa* | 1 | + |
| 2 | 45/f | *S. epidermidis* | 1 | – |
| 3 | 58/f | *S. epidermidis, Acinetobacter lwoffii* | 2 | + |
| 4 | 63/m | *Enterobacter* sp., *Neisseria sicca, Corynebacterium hoffmannii, Streptococcus viridians* | 4 | + |
| 5 | 44/m | *S. epidermidis, K. pneumoniae* | 2 | + |
| 6 | 78/m | *Escherichia coli* | 1 | + |
| 7 | 64/f | NG | - | + |
| 8 | 81/m | *Alcaligenes faecalis* | 1 | + |
| 9 | 67/m | NG | - | + |
| 10 | 54/m | NG | - | + |
| 11 | 54/f | NG | - | – |
| 12 | 37/f | NG | - | + |
| 13 | 59/f | *Haemophilus influenzae* | 1 | – |
| 14 | 13/f | NG | - | + |
| 15 | 59/m | NG | - | + |
| 16 | 27/m | *A. lowffii* | 1 | + |
| 17 | 60/f | NG | - | – |
| 18 | 39/m | *Enterobacter* sp., *H. influenzae* | 2 | + |
| 19 | 48/m | NG | - | + |
| 20 | 52/f | NG | - | – |

[a]In years.

[b]m, male; f, female.

[c]NG, No growth.

[d](+), positive; (–), negative.

to the predominant bacterial species that commonly found in clinical studies on CRS [32–34] and displayed a biofilm-positive phenotype on the surface tissue from CRS patients in this study. *P. aeruginosa* ATCC 27853 was also included in this study as reference strain. Antimicrobial susceptibility test results of the bacterial isolates are shown in S1 Table. *K. pneumoniae* and *P. aeruginosa* were regularly cultured on MAC (Himedia®) while *S. epidermidis* was grown on Nutrient agar (NA) (Himedia®). Plates were incubated at 37˚C for 24 h. A single colony of *K. pneumoniae* and *P. aeruginosa* were cultured aerobically in modified Vogel and Bonner's medium (MVBM) at 37˚C in a 200 rpm shaker incubator for 18 h. *S. epidermidis* was cultured overnight in brain heart infusion (BHI) (Himedia®) under the same conditions as mentioned above.

## Antibiotic susceptibility testing

MICs and MBCs of conventional antibiotics were determined using a broth microdilution technique [35]. Briefly, two-fold serial dilutions of AMX (Sigma-Aldrich, St. Louis, MO) were prepared in Mueller Hinton broth (MHB) (Himedia®) at concentrations ranging from 17 to 34209 µM (6–12500 µg/ml) in 96-well microtiter plates (Nunclon™, Roskilde, Denmark) for antimicrobial susceptibility testing. Only *P. aeruginosa* was additionally tested for susceptibility to TOB (Sigma-Aldrich) (0.03–52 µM or 0.01–25 µg/ml). Bacterial suspensions ($5 \times 10^5$ CFU/ml) were added to the microtiter plates, which were then incubated at 37˚C for 24 h. Antibiotic-free culture was used as a control. After incubation, MIC values were observed according to the guidelines of the Clinical & Laboratory Standards Institute (CLSI). The results were confirmed using a microplate reader at $OD_{620}$ (Sunrise™, TECAN, Australia). The plate count technique was further used to determine the MBC values on Mueller Hinton agar (MHA) (Himedia®). Each experiment was performed independently three times in triplicate.

## Antimicrobial activity of antibiotics and peptides against planktonic bacteria

Each bacterial suspension ($5 \times 10^5$ CFU/ml) in 1 mM potassium phosphate buffer (PPB) (pH 7.0) was added to an equal volume of each antibiotic (AMX and TOB) and AMP (LL-37, LL-31, D-LL-37 and D-LL-31) to reach final concentrations of 1, 5 and 10 µM and incubated at 37˚C for 1 and 2 h. A bacterial suspension without peptide served as a control. The samples were then serially diluted and cultivated on NA (Himedia®) and incubated for 24 h to allow colony counting. The percentage killing by each AMP was calculated using the formula [1 − (CFU sample/CFU control)] × 100% [15, 36]. Each experiment was performed independently three times in duplicate.

## Determination of the biofilm-forming capacity

The biofilm-forming capacity of each strain was examined using the biofilm crystal violet staining assay (on 2-day biofilm) as previously described [37]. Two hundred microlitres of bacterial suspension ($10^8$ CFU/ml) were added into a sterile 96-well flat-bottom plate (Nunclon™) and the plate was incubated at 37˚C for 3 h. Cell-free medium was used as a control. Weakly adhered planktonic cells were removed and fresh medium was added, then the plate was incubated at 37˚C for 21 h. The biofilms were washed with sterile distilled water, fresh medium was added and the plate was incubated at 37˚C for 24 h. The adherent biofilms were then fixed with 99% (v/v) methanol, air dried and stained with 2% (w/v) crystal violet for 5 min. Subsequently, 2-day biofilms were solubilized with 33% (v/v) glacial acetic acid. The absorbance was spectrophotometrically measured at the wavelength of 620 nm using a microplate reader (Sunrise™). Each experiment was performed independently three times in 8 replicates.

## Comparison of the antimicrobial activity of peptides and antibiotics against bacteria grown as biofilm

The effects of both conventional antibiotics and AMPs against bacteria under biofilm-stimulating condition were determined using a transferable solid phase (TSP) pin lid, which resembles the 'Calgary' biofilm device as previously described with some modifications [38]. Each strain ($10^7$ CFU/ml) was added to sterile 96-well plates (Nunclon™), then covered by a TSP pin lid (NUNC™, Roskilde, Denmark), and further incubated at 37˚C for 24 h. Following the period of incubation, biofilms formed on the TSP pin surfaces were washed by sterilized distilled water and challenged with AMX in a range from 134 to 34418 μM (50–25000 μg/ml) and 0.05 to 100 μM of LL-37, LL-31, D-LL-37 and D-LL-31 in 1 mM PPB. Additionally, the effect of TOB (2–836 μM or 1–391 μg/ml) against *P. aeruginosa* biofilm was also investigated. Then the plates were statically incubated at 37˚C for 24 h. A well free of antimicrobial agent was used as a control. After 24 h, bacterial survival was determined using the plate count technique on NA (Himedia®). The percentage killing of each antimicrobial agent was calculated using the formula [1 − (CFU sample/CFU control)] × 100% [15, 19]. The concentration of each agent required to kill 50% of cells in each tested bacterial strain ($IC_{50}$ value) was determined by linear regression of plots. Each experiment was performed independently three times in duplicate.

## The synergistic activity of D-LL-31 with antibiotics against bacteria grown as biofilm

After the $IC_{50}$ value of each antimicrobial agent was calculated, D-LL-31 (the most effective peptide) was then selected to determine its synergistic activity with conventional antibiotics (AMX and TOB) using the broth microdilution checkerboard technique with some modification [15, 39]. Briefly, 24 h biofilms on TSP pin lids were mixed with the antibiotics (AMX or TOB) and D-LL-31 diluted in 1 mM PPB to final concentrations of $IC_{50}$ down to 1/16 of the $IC_{50}$ values of each agent and incubated at 37˚C for 24 h (Table 4). Bacterial viability in each interaction was determined using the plate count technique. The fractional inhibitory concentration index (FICI) values were calculated and interpreted in terms of synergism/antagonism as previously described [40]. Each experiment was performed independently three times in duplicate.

## The effect of D-LL-31 alone and in combination with antibiotics on biofilm matrix

The *in vitro* Amsterdam Active Attachment Model (AAA-model) was used to determine the effect of peptides alone and in combination with conventional antibiotics on biofilm matrix [41]. Briefly, 1.5 ml of each overnight culture ($10^8$ CFU/ml) was added to a 24-well tissue culture plate (Nunclon™). The plate was covered with a sterile stainless-steel lid containing glass coverslips and aerobically incubated at 37˚C for 24 h. After growing the biofilm, D-LL-31 or AMX alone ($IC_{50}$ values; 0.6 μM D-LL-31 and 7925 μM AMX (2896 μg/ml)) and a combination of both (based on the synergy data; 0.07 μM D-LL-31 and 495 μM AMX (181 μg/ml)) were used in *K. pneumoniae* biofilm matrix challenge. For *S. epidermidis* biofilm matrix, $IC_{50}$ values of D-LL-31 (0.5 μM) or AMX (33092 μM or 12092 μg/ml) and combination of D-LL-31/AMX (0.03 μM D-LL-31 and 2068 μM AMX (756 μg/ml)) were used. For *P. aeruginosa* biofilm matrix, $IC_{50}$ values of D-LL-31 (4 μM) or AMX (45163 μM or 16502 μg/ml) or TOB (4 μM or 2 μg/ml) and combination of D-LL-31/AMX (0.25 μM D-LL-31 and 2823 μM AMX (1031 μg/ml)) or D-LL-31/TOB (0.5 μM D-LL-31 and 0.2 μM TOB (0.1 μg/ml)) were used. All

tested conditions were incubated at 37°C for 24 h. A well free of antimicrobial agent was used as a control. Following incubation, biofilms were fixed with 2.5% glutaraldehyde (Sigma-Aldrich) at 25°C for 3 h. The fixed bacterial biofilms on glass coverslips were stained with 50 μg/ml Alexa Fluor® 488 (Invitrogen, Carlsbad, CA) for 30 min. This stain reacts with exo-polysaccharide matrixes of the biofilm. The glass coverslips were then mounted with 80% glycerol and the stained cells were photographed using a confocal laser-scanning microscope (CLSM) (TCS SP8-Leica Microsystems, Wetzlar, Germany). The biofilm biomass was used as a quality-control parameter to confirm the reduction of biofilm matrix using COMSTAT analysis (BioCentrum-DTU, Lungby, Denmark). Each experiment was performed independently three times in duplicate.

## The localization of peptide on biofilm matrix

The localization of TAMRA-labeled D-LL-31 on biofilm matrix was investigated using the AAA-model as mentioned above. One-day biofilm was fixed with 2.5% glutaraldehyde (Sigma-Aldrich) at 25°C for 3 h and stained with 50 μg/ml of Alexa Fluor® 488 (Invitrogen) for 30 min. Subsequently, TAMRA-labeled D-LL-31 at the $IC_{50}$ concentration was added onto the biofilm architecture of each tested strain for 5 min, and then the biofilm was washed 3 times using sterile distilled water. The localization of D-LL-31 on biofilm matrix was visualized by CLSM (TCS SP8-Leica Microsystems). Each experiment was performed independently three times in duplicate.

## Determination of the cytotoxicity of D-LL-31 for A549 cells

Human lung epithelial cells (A549, ATCC®, CCL-185™) were used to investigate the cytotoxicity of D-LL-31 as previously observed [42]. Briefly, A549 cells were maintained in RPMI-1640 (HyClone™, Marlborough, MA) with 10% fetal bovine serum (FBS) (HyClone™) and sub-cultured by standard methods using trypsin/EDTA (0.25% trypsin, 0.1% EDTA) (Corning™, Corning, NY). A standard curve of A549 cells (100–50000 cells/well) was constructed as previously described with some modification [43]. For MTT assay, cells were plated at a density of $2.5 \times 10^4$ cells/well in 96-well plates and grown overnight. D-LL-31 was diluted in PPB and added to culture medium to final concentrations ranging from 0.05 to 100 μM. Prepared A549 cells were then exposed to the diluted peptides for 24 h at 37°C, 5% $CO_2$ atmosphere. Negative controls received either culture medium or PPB. Cells with 2% dimethyl sulfoxide (DMSO) (Sigma-Aldrich) were used as positive control. Cytotoxicity was assessed using MTT (0.5 mg/ml) according to the manufacturer's instructions (Sigma-Aldrich). Absorbance was measured at a wavelength of 570 nm with background subtraction at 630–690 nm. Then absorbance values from the MTT were converted to cells/well using a standard curve. Cell viabilities were calculated using the formula; [mean of cells in treated well]/[mean of cells in control well] × 100%. Each experiment was performed independently three times in triplicate.

## Detection of cytokine release from D-LL-31-treated A549 cells

The immunomodulatory effect of D-LL-31 on A549 human lung epithelial cells was investigated using cytokine antibody arrays. A total of $5.5 \times 10^6$ A549 cells were seeded into a T25 tissue culture flask (Corning™) and grown overnight at 37°C, 5% $CO_2$. The cells were incubated with 10 μM of D-LL-31 at 37°C, 5% $CO_2$ for 24 h (This concentration spanned the $IC_{50}$ values of the peptide against all tested bacterial strains grown as biofilm and showed no cytotoxicity in A549 cells). Cells in medium without peptide were used as a control. After incubation, the secreted cytokines in culture supernatant were detected using a human angiogenesis antibody array kit (AAH-ANG-1000-4, Norcross, GA) according to the manufacturer's instructions.

The images were taken using a chemiluminescence imaging system and the densitometry data were calculated using ImageQuant TL Software version 8.1 (Amersham™ Imager 600, GE Healthcare Life Sciences, Chicago, IL). Each experiment was performed independently two times in duplicate.

## Statistical analysis

The percentage of killing activity of all tested agents against bacteria grown as biofilm, biofilm biomass, cytotoxicity and densitometry data of cytokine expression are presented as mean ± standard deviation (SD). Comparisons between the average percentage killing activity of each AMP at the same concentration in all tested agents were analyzed using one-way ANOVA. The statistical significance of cell viabilities and pixel density ratio in each treatment compared with controls was determined by Student's t test. All statistical testing was performed using the SPSS software, version 16.0 (Chicago, IL).

## Results

### Isolation and identification of bacteria from CRS patients

Characteristics of the 20 patients (ten males and ten females) with CRS are shown in Table 2. The ages ranged from 13 to 81 years (average 53 years). None was pregnant or had underlying diseases; diabetes mellitus, cancer or AIDS. Of 20 clinical specimens, ten yielded positive cultures on at least one of the media that were used. Sixteen microorganisms were isolated from maxillary or ethmoid sinus washings, 11 of which (69%) were Gram-positive and 5 were Gram-negative bacteria. About 75% (12/16) of the isolates were classified as facultative anaerobic bacteria and 25% (4/16) as obligate aerobes or aerobic bacteria.

### The presences of biofilms on sinus mucosal tissues of CRS patients

Investigation using SEM revealed evidence of biofilm formation in about 75% (15/20) of sinus mucosal tissues samples from CRS patients, as shown in Table 2. Scanning electron micrographs of mucosal tissues with the indication of biofilms are shown Fig 1. The presence of a biofilm was defined on the basis of visible bacterial aggregation located on the surface of the epithelium. Large biofilm aggregates were observed (Fig 1C and 1D) and are distinct from the appearance of ciliated epithelium from samples that yielded bacteria in culture but without evidence of biofilm (Fig 1A). Coccus- and bacillus-shaped elements were observed on the biofilm structure (Fig 1D, white arrow). Biofilm obtained from CRS mucosal tissues showed strands of extracellular materials between the cells, which might represent extracellular DNA (eDNA) from bacterial and human cells (Fig 1B).

### Antimicrobial activity of antibiotics and AMPs against all tested bacteria in planktonic form

The MIC and MBC values of conventional antibiotics against bacteria isolated from CRS patients are shown in Table 3. Resistance to AMX was observed in all strains. *P. aeruginosa* in planktonic form was susceptible TOB. All tested peptides, at 1 μM, exhibited ≥ 95% killing activity against all tested isolates after 1 h of incubation (Fig 2). All peptides at 5 μM completely killed the initial inoculum after 2 h. All concentrations of AMX showed a low killing ability (about 0.5–18.5%) toward all bacterial strains in planktonic form during 1 and 2 h of incubation. TOB at concentrations of 1 and 5 μM showed ≥ 92% killing effect against the clinical isolate *P. aeruginosa* at 1–2 h, but 10 μM was required to kill all bacterial cells (Fig 2C) while only 1 μM TOB showed ≥ 98% killing effect against the reference strain *P. aeruginosa* ATCC 27853

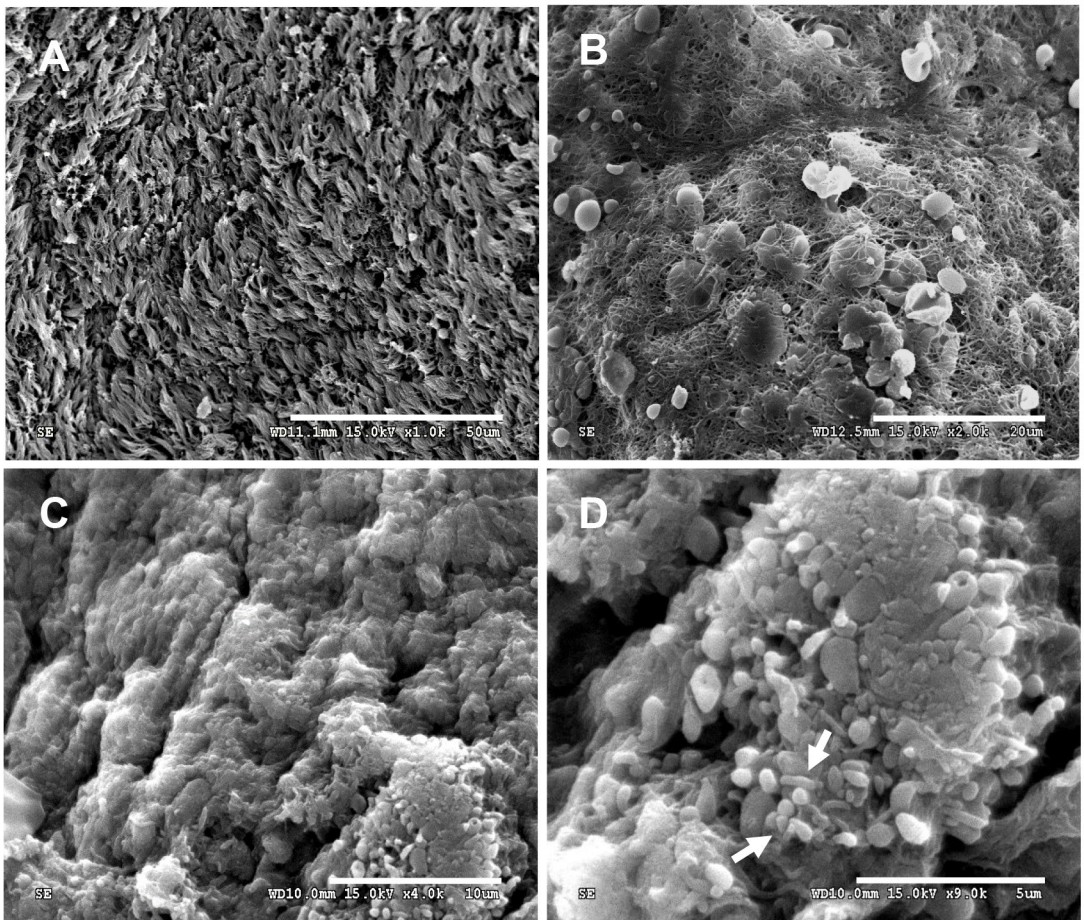

**Fig 1. Scanning electron micrographs of mucosal tissues from CRS patients.** (A) Maxillary sinus mucosa without evidence of biofilm but positive for bacteria when cultured; bar = 50 μm. (B) Wire-like structures seen on the sinus mucosal epithelium; bar = 20 μm. (C) Large biofilm aggregates; bar = 10 μm and (D) biofilm with coccus- and bacillus-shaped elements (white arrows); bar = 5 μm.

at 1 h and both 5 and 10 μM killed all bacterial cells within 1 h (Fig 2D). Interestingly, only D-LL-31 (1 μM) caused rapid killing of all tested strains: bacterial cells at 5 x $10^5$ CFU/ml were completely killed within 1 h of the peptide treatment.

## The quantification of bacterial biofilm formation

The crystal violet assay (using 2-day biofilm) was performed to determine the biofilm-formation capacity of each bacterial stain. *P. aeruginosa* showed the greatest tendency for biofilm

**Table 3. Antimicrobial susceptibility of antibiotics against bacterial isolated from CRS patients in planktonic form.**

| Isolates | Antibiotics | MIC (μM/μg/ml) | MBC (μM/μg/ml) |
|---|---|---|---|
| *K. pneumoniae* | AMX | 4276/ 1562.5 | 8552/ 3125 |
| *S. epidermidis* | AMX | 4276/ 1562.5 | 17104/ 6250 |
| *P. aeruginosa* | AMX | 17105/ 6250 | 34209/ 12500 |
| | TOB | 0.2/ 0.1 | 1/ 0.4 |

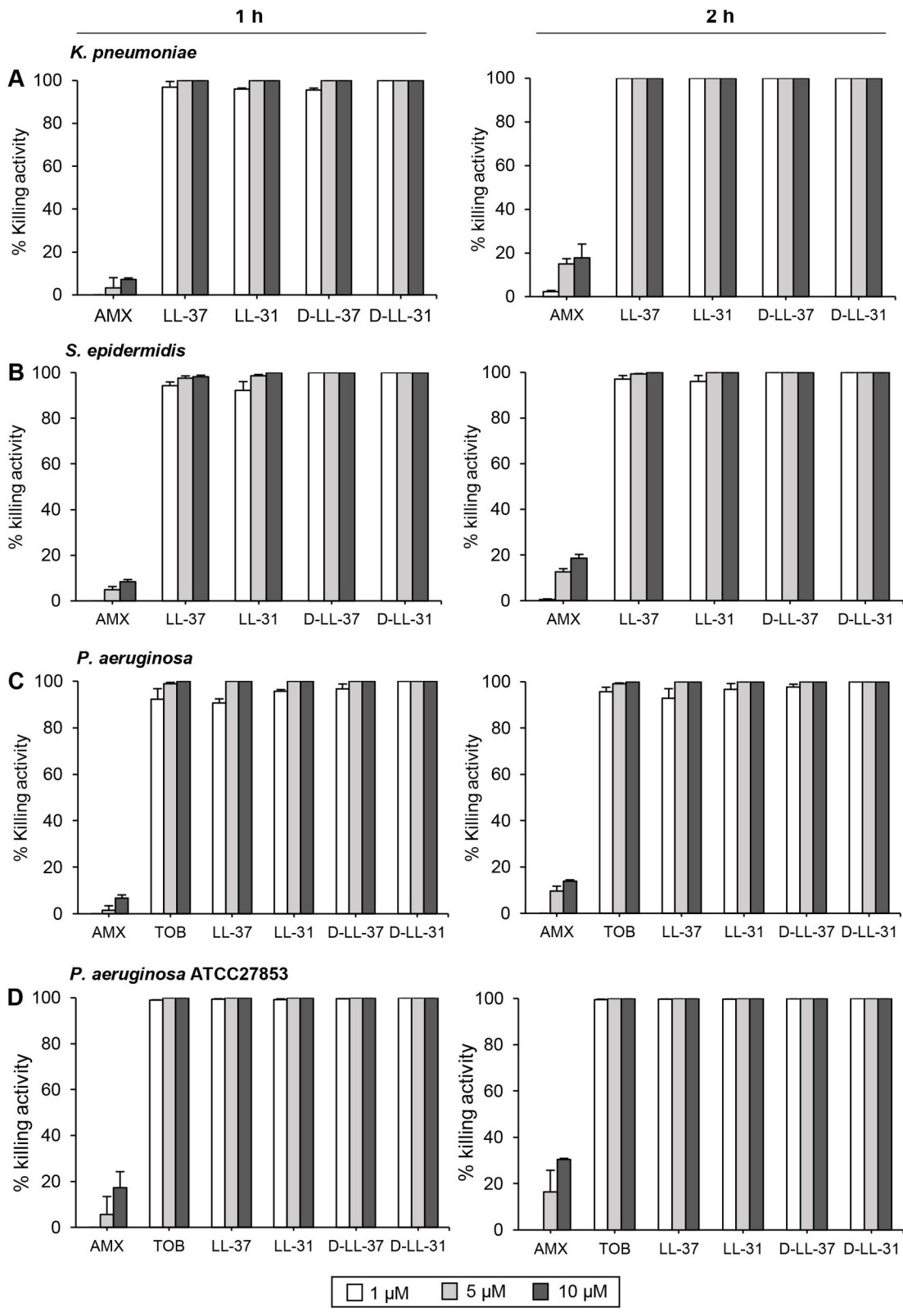

**Fig 2. Antimicrobial activity of antibiotics and AMPs against planktonic culture.** Bacterial suspensions of (A) *K. pneumoniae*, (B) *S. epidermidis* and (C) *P. aeruginosa* (D) *P. aeruginosa* ATCC 27853 were incubated with 1, 5 and 10 μM of each peptide for 1 and 2 h. The results are presented as mean values ± SD of three independent experiments, carried out in duplicate.

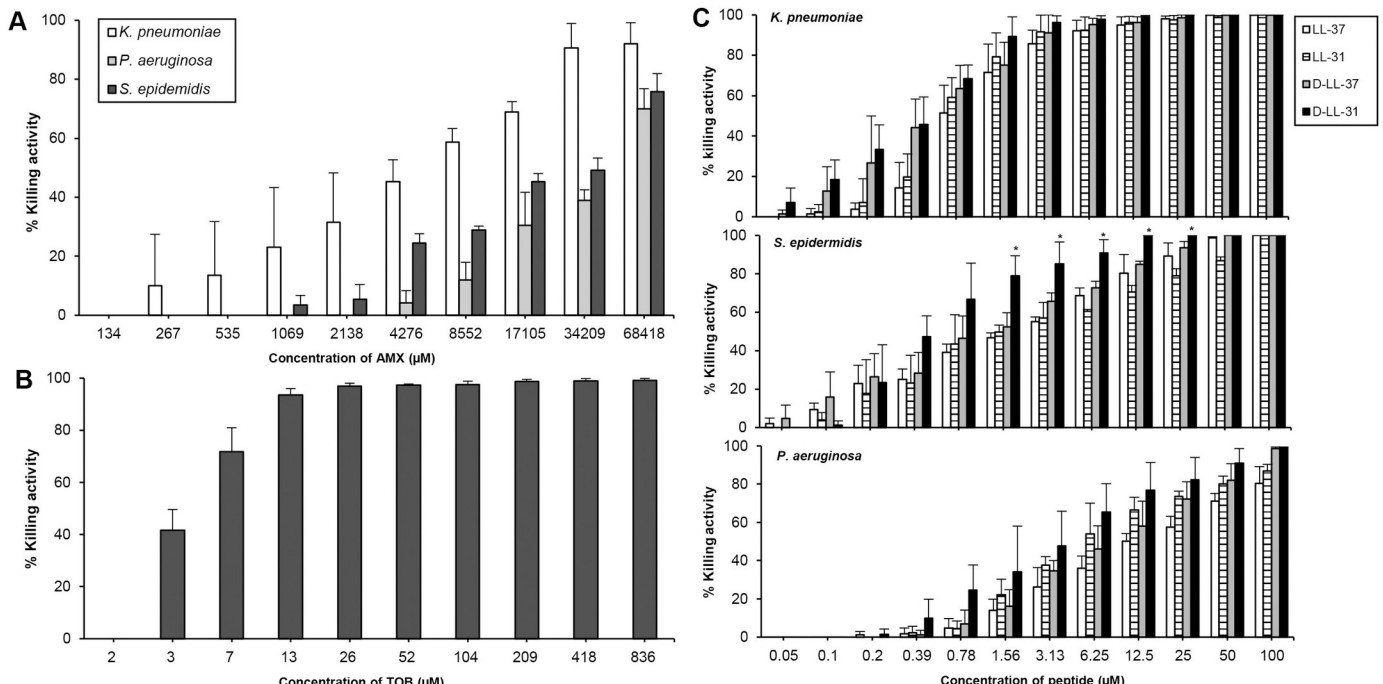

**Fig 3. The killing activity of AMPs and antibiotics against all tested bacterial strains grown as biofilm.** (A) Effect of AMX against all tested isolates and (B) TOB toward only *P. aeruginosa* biofilm. (C) The killing capacity of AMPs against all tested bacteria. The 24 h biofilm of each bacterial isolate was incubated with AMPs and antibiotics for 24 h. The results are presented as mean values ± SD of three independent experiments, carried out in duplicate. One-way ANOVA was used for determining the statistical significance of the killing activities of each antimicrobial peptide in different concentrations (*$P<0.05$ comparing between each antimicrobial peptide at the same concentration).

formation [$OD_{620}$ (mean ± SD); 0.376 ± 0.1], followed by *S. epidermidis* and *K. pneumoniae*, 0.273 ± 0.05 and 0.203 ± 0.04, respectively. Therefore, the biofilm-producing capacity of all bacterial strains was confirmed.

## Killing effect of all tested agents against bacteria under biofilm-stimulating condition

All agents displayed a clear dose- and strain-dependent killing activity against all tested bacteria grown as biofilm (Fig 3). The highest concentration of AMX (68418 μM or 25000 μg/ml) exhibited 70%, 75%, and 90% killing activity against *P. aeruginosa*, *S. epidermidis* and *K. pneumoniae* biofilm, respectively (Fig 3A). Biofilm of *P. aeruginosa* was found to be sensitive to TOB (Fig 3B). Among all peptides, D-LL-31 exhibited the strongest killing activity against all bacterial strains (Fig 3C). The killing activities of D-LL-31 at concentrations 1.56–25 μM were significantly higher than all tested peptides ($P < 0.05$) against *S. epidermidis* biofilm. However, there were no statistically significant differences in killing activity of D-LL-31 against biofilm of *K. pneumoniae* and *P. aeruginosa* when compared to the other peptides.

The $IC_{50}$ values of each agent was further investigated (Table 4). D-LL-31 exhibited the lowest $IC_{50}$ values against all bacteria in biofilm at concentration ranging from 0.5 to 4 μM. $IC_{50}$ values of AMX against biofilm of *P. aeruginosa*, *K. pneumoniae*, and *S. epidermidis* were higher than those of D-LL-31 (about 11000-, 13000- and 66000-fold, respectively). In contrast, the $IC_{50}$ values of TOB and D-LL-31 toward *P. aeruginosa* biofilm were similar. However, these findings also indicated that D-LL-31 was more effective than the L-form and the LL-37 variants against all bacteria grown as biofilm.

**Table 4. The IC$_{50}$ values of antimicrobial peptides and antibiotics against bacterial isolated from CRS patients under biofilm-stimulating condition.**

| Isolates | IC$_{50}$ value (µM) | | | | | |
|---|---|---|---|---|---|---|
| | LL-37 | LL-31 | D-LL-37 | D-LL-31 | AMX | TOB[a] |
| *K. pneumoniae* | 0.8 | 0.7 | 0.8 | 0.6 | 7925 (2896 µg/ml) | ND |
| *S. epidermidis* | 2.4 | 1.8 | 1.3 | 0.5 | 33092 (12092 µg/ml) | ND |
| *P. aeruginosa* | 11.5 | 5.3 | 10.1 | 4 | 45163 (16502 µg/ml) | 4 (2 µg/ml) |

[a]ND, not determined.

## Synergistic effect of D-LL-31 with antibiotics

The most effective antimicrobial peptide, D-LL-31, showed synergistic interactions with all tested antibiotics against bacteria grown as biofilm based on FICI ≤ 0.5 (Table 5). The combination of D-LL-31 with antibiotics exhibited 50% killing against each bacterial strain at IC$_{50}$ values that were at least 8- to 16-fold lower than those of either agent alone. A 16-fold reduction in IC$_{50}$ values of both D-LL-31 and AMX were found against *P. aeruginosa* and *S. epidermidis* biofilm (FICI = 0.125). The interaction of D-LL-31 with AMX and TOB reduced the IC$_{50}$ values of the antibiotics by 8- to 16-fold against *K. pneumoniae* and *P. aeruginosa* biofilm, respectively (FICI = 0.188).

## Effect of D-LL-31 alone and in combination with antibiotics against biofilm matrix under static conditions

The green fluorescence of Alexa Fluor® 488 was used to represent the biofilm matrix of all tested bacteria after exposure to D-LL-31 alone or in combination with antibiotics (Fig 4). A combination of D-LL-31 with AMX led to a distinct decrease in biofilm matrix for all tested strains relative to controls. A statistically significant reduction of *P. aeruginosa* biofilm matrix occurred when treated with a combination of D-LL-31 and TOB (Fig 4C). Moreover, a greater reduction of biofilm matrix was observed in all tested strains after challenge with D-LL-31 alone than after challenge with antibiotic alone.

To confirm the biofilm-disrupting effect of the agents, biofilm biomass (µm$^3$/µm$^2$) was measured. Significant decreases of biomass were observed following D-LL-31 treatment alone ($P < 0.01$) and in combination with antibiotics ($P < 0.001$) relative to controls. There were no statistically significant effects of either antibiotic tested against *K. pneumoniae* and *P. aeruginosa* biofilm. However, AMX had a significant effect on *S. epidermidis* ($P < 0.05$).

## Localization of D-LL-31 on biofilm matrix

The localization of the peptide on biofilm architecture was observed using TAMRA-labeled D-LL-31. Image slices were obtained at 2-µm increments moving from the top of the biofilm

**Table 5. The synergistic interaction of D-LL-31 with conventional antibiotics against bacteria in biofilm condition.**

| Isolates | Antimicrobial agent in concentration of combination (µM) | | | FICI values | Fold decrease of IC$_{50}$ values[b] | | |
|---|---|---|---|---|---|---|---|
| | D-LL-31 | AMX[a] | TOB[a] | | D-LL-31 | AMX[a] | TOB[a] |
| *K. pneumoniae* | 0.07 | 495 (181 µg/ml) | ND | 0.188 | 8 | 16 | ND |
| *S. epidermidis* | 0.03 | 2068 (756 µg/ml) | ND | 0.125 | 16 | 16 | ND |
| *P. aeruginosa* | 0.25 | 2823 (1031 µg/ml) | ND | 0.125 | 16 | 16 | ND |
| | 0.5 | ND | 0.2 (0.1 µg/ml) | 0.188 | 8 | ND | 16 |

[a]ND, not determined.
[b]Fold decrease of each antimicrobial agent were calculated from their IC$_{50}$ value compared with concentration of combination (µM).

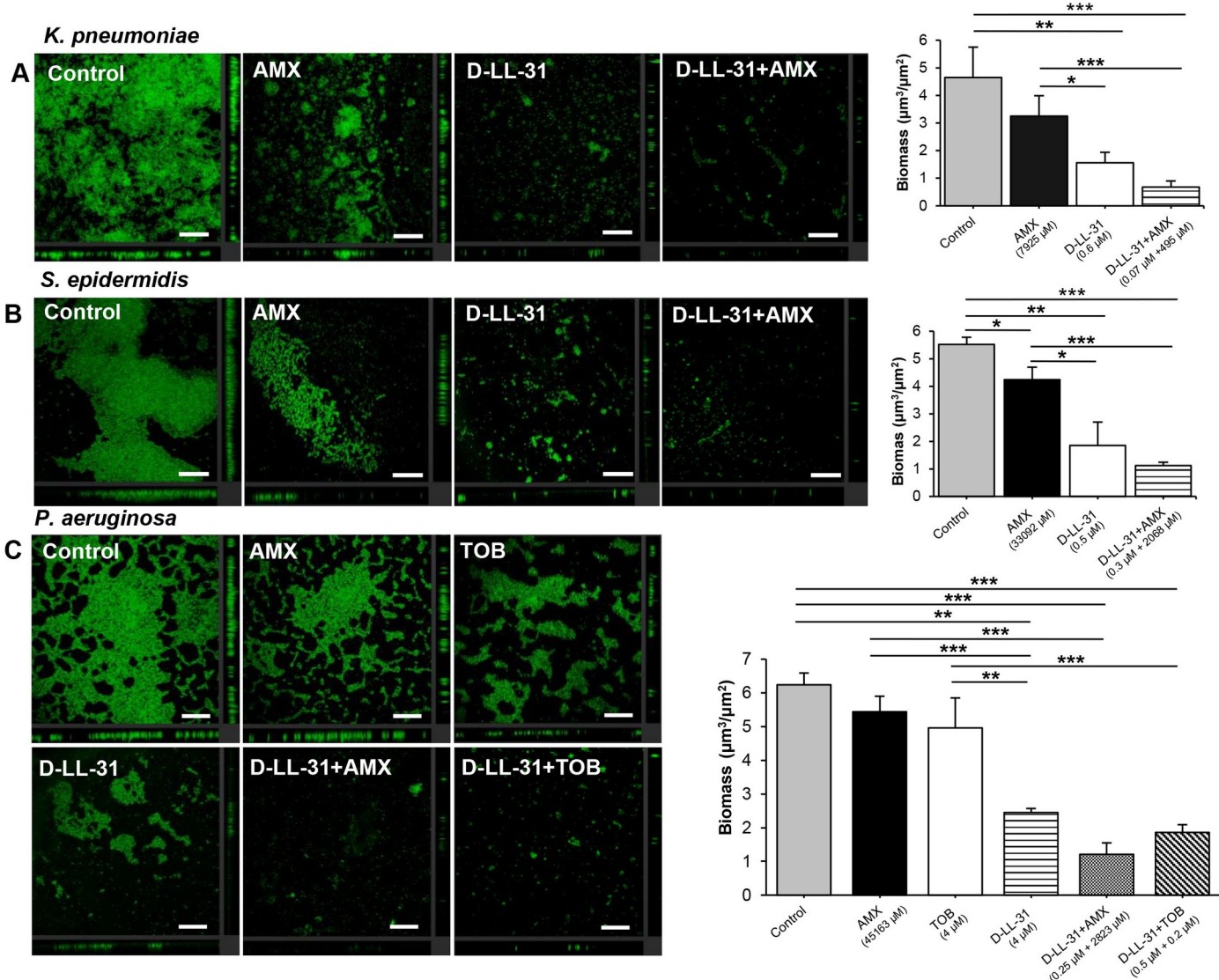

**Fig 4. Effect of D-LL-31 alone and in combination with antibiotics on biofilm matrix.** The 24 h biofilm of (A) *K. pneumoniae*, (B) *S. epidermidis* and (C) *P. aeruginosa* were incubated with D-LL-31 alone and in combination with antibiotics (AMX or TOB) for 24 h. Biofilm were stained with Alexa Fluor® 488. Biomass was analyzed using COMSTAT and values are shown as mean ± SD in duplicate from three independent experiments. *$P<0.05$, **$P<0.01$ and ***$P<0.001$. Scale bar indicate 100 μm.

matrix through the substrate. The stacked sections of biofilm matrix of all tested bacterial strains in the presence of peptide are shown in Fig 5. A rapid binding of D-LL-31 to biofilm matrix of all tested bacteria was observed within 5 min of incubation. D-LL-31 could be detected in all stacks of *K. pneumoniae* and *S. epidermidis* biofilms (Fig 5A and 5B). The peptide accumulated mainly in the top layers of *P. aeruginosa* biofilm (Fig 5C), but some could also be seen in the middle layers.

## Cytotoxicity of D-LL-31 on human lung epithelial cells

A cell-survival assay was performed to determine the cytotoxicity of D-LL-31 on human lung epithelial cells as shown in Fig 6. More than 90% of cells remained viable after treatment with 12.5 μM D-LL-31, and more than 80% after treatment with 25 μM, indicating that D-LL-

|  | Upper | Middle | Lower |
|---|---|---|---|

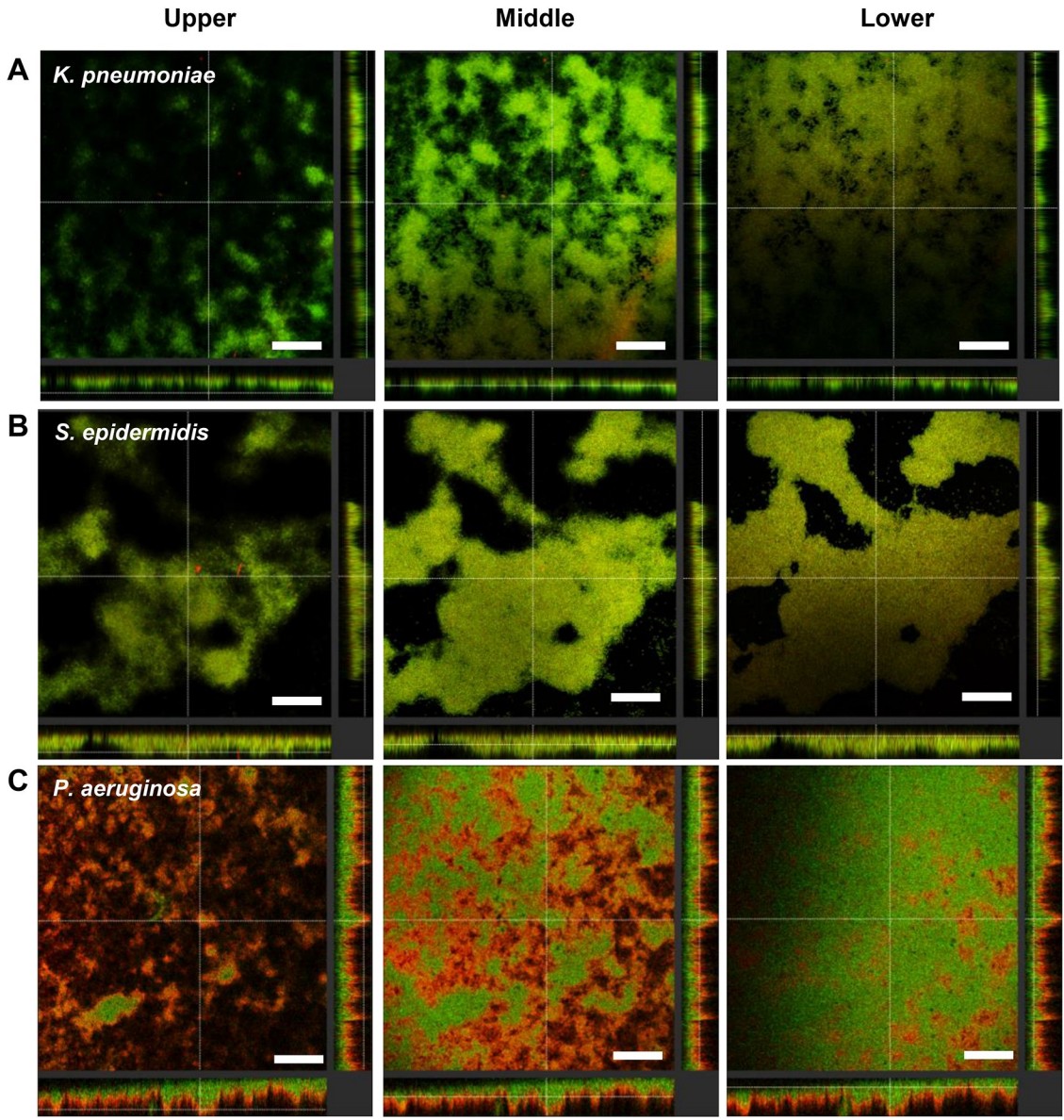

**Fig 5. The localization of TAMRA-labelled D-LL-31 on biofilm matrix.** The 24h biofilm of (A) *K. pneumoniae*, (B) *S. epidermidis* and (C) *P. aeruginosa* were stained with Alexa Fluor® 488 (green) then incubated with TAMRA-labeled D-LL-31 (red) for 5 min. Each z-stack image was visualized using CLSM. Scale bar indicate 100 μm.

31exhibited low cytotoxicity against A549 cells at concentrations that are effective against the bacteria grown as biofilm (IC$_{90}$ values of D-LL-31 at concentration ranging from 1 to 25 μM). However, treatment with $\geq$ 50 μM D-LL-31 resulted in more than 90% cytotoxicity of A549 cells.

## Expression of cytokines and angiogenesis-related proteins

The expression of angiogenesis-related proteins and inflammatory cytokines in the presence of D-LL-31 was determined using a cytokine array membrane assay. The pixel density ratio of each identified protein is shown in Fig 7A and 7B. Epithelial-derived neutrophil-activating peptide 78 (ENA-78) was significantly secreted in the presence of D-LL-31 ($P < 0.05$) relative

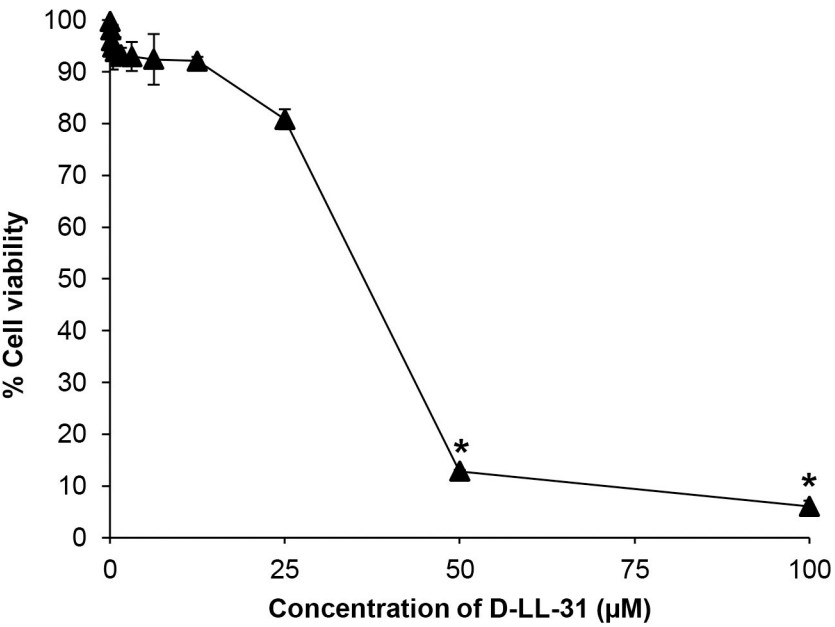

**Fig 6. Cytotoxicity of D-LL-31 on human lung epithelial cells.** The cells were incubated with different concentrations of D-LL-31 (0.05–100 µM) for 24 h. The cell viability was assessed using the MTT assay. Data are presented as the mean ± SD (*$P<0.001$).

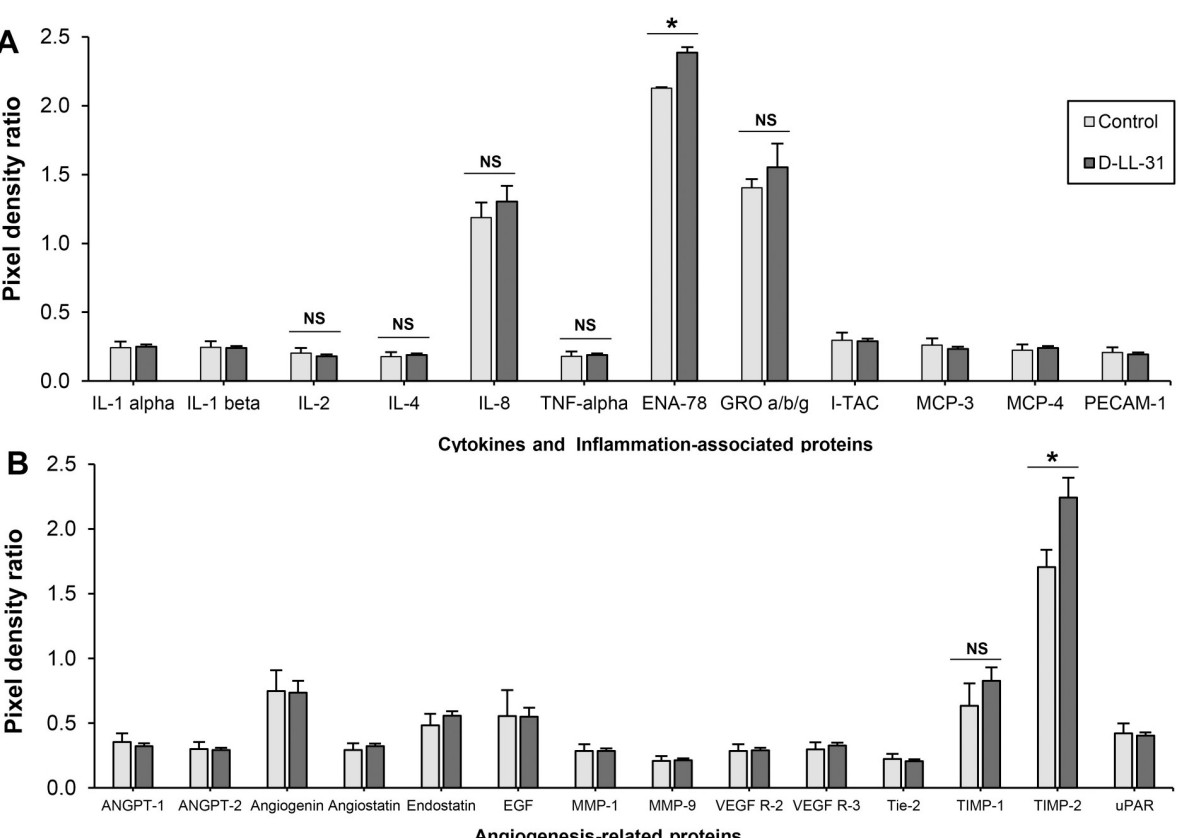

**Fig 7. Expression of cytokines and angiogenesis-related proteins in the presence of D-LL-31.** The cells were incubated with 10 µM D-LL-31 for 24 h. (A) The cytokine and (B) angiogenesis-related protein were investigated using cytokine antibody arrays. The pixel density ratio of each identified proteins was calculated using ImageQuant TL Software and presented as mean ± SD. (*$P<0.05$).

to controls (Fig 7A). There was no significant change in levels of cytokines that might impact CRS, such as IL-4, IL-8, TNF-α and growth-related oncogene-a, b and g (GRO-a, b and g). The expression of IFN-γ, an important cytokine in the immunopathogenesis of CRSsNP, was not found after incubation with D-LL-31. Moreover, treatment with D-LL-31 did not induce expression of IL-6, a marker of inflammation in CRSwNP. In addition, TIMP-2 was significantly elevated in culture supernatant with D-LL-31 ($P < 0.05$) (Fig 7B).

## Discussion

Bacterial infection play an important role in the pathogenesis, cause inflammation and prescribe antimicrobial therapy in CRS [44, 45]. Moreover, biofilm on mucosal tissues of CRS patients have demonstrated the significant role in the pathogenesis and disease progression of CRS [46] and its association with antibiotic resistance [10, 11]. In this study, the enhancement of biofilm-eradicating effect of currently used antibiotics for CRS treatment with cathelicidin-derived peptides and D-enantiomers were observed. Various bacterial species were isolated from maxillary and ethmoid sinus irrigation samples from CRS patients. Of these, *S. epidermidis* (a colonizer of the nasal cavity) was the most common, in agreement with previous reports [33]. Moreover, Gram-negative bacilli such as *Enterobacter* spp, *E. coli*, *K. pneumoniae* and *P. aeruginosa* which are considered as agents of CRS infection [47, 48] were also found in this study. The SEM micrographs revealed that 75% (15/20) of CRS patients had evidence of biofilms on their nasal mucosa, consistent with other studies [7, 33]. Seven of the 15 individuals that were biofilm-positive did not yield bacteria in culture. In these cases, biofilm might be a product of anaerobic bacteria and fungi. Anaerobic organisms are relatively uncommon in the nasal cavity [44] and fungi are found in only 1.6% of CRS patients [47]. Therefore, these groups of microorganisms were excluded in this study. In addition, using standard culture-based methods was the limitation in this study which may affect the exact number of isolated bacteria from CRS patients. Molecular technique should be used as addition tool to identify microorganisms in clinical samples. Finally, *K. pneumoniae*, *P. aeruginosa* and *S. epidermidis* were selected according to the known prevalence of these species in CRS patients with biofilm [47, 48] for further experiments.

AMX is a reasonable first-line antibiotic for treatment of sinus infections in many geographic areas [49]. In addition, TOB is recommended to treat *P. aeruginosa* infection [50]. Our antimicrobial susceptibility results showed that all bacterial strains isolated from CRS patients in planktonic form were resistant to AMX, as previously reported [51]. Despite their resistance to AMX, we observed that the bacteria were sensitive to all AMPs tested. We were able to demonstrate that the D-enantiomeric LL-31 peptide had a higher antibacterial potency against all tested bacteria in planktonic culture than the other AMPs (LL-37, LL-31 and D-LL-37) and currently used antibiotics (AMX and TOB).

Long-term antibiotic therapy in CRS patients may be one of the reasons of the antibiotic-resistance. Moreover, microbial biofilm can be up to 1000 times more resistant to antibiotic treatment than its planktonic form in CRS patients [34, 52]. Thus, we investigated the killing capacity of the tested agents against bacteria grown as biofilm. Our findings are consistent with Kifer D et al [53], given the greater resistance to AMX than their planktonic counterparts. However, all bacterial isolates grown as biofilm tended to be more sensitive to all peptides, especially D-LL-31. This protease-resistant peptide displayed a killing activity comparable to that of TOB against *P. aeruginosa* biofilm and greater biofilm-reduction properties than TOB. The mode of action of AMPs relies on their capacity to disrupt bacterial membranes, even in slow-growing or dormant biofilm-forming cells [8, 54]. Thus, AMPs are better able to combat bacterial cells during biofilm-stimulating conditions. Several studies have shown that AMPs

may penetrate into the biofilm matrix and kill the persister cells, a great advantage over conventional antibiotics [8, 15].

The biofilm matrix accounts for up to 90% of the total biofilm mass and acts as a barrier for antibiotic diffusion [55], which largely explains why bacteria in biofilms are so resistant to antibiotics [8]. We have shown previously that D-LL-31 may disrupt the biofilm matrix leading to enhanced access by ceftazidime to kill bacterial cells, ultimately resulting in synergistic interaction [15]. Here, TAMRA-labelled D-LL-31 was used to observe the localization of the peptide before it exerted its effect. D-LL-31 was observed in all layers of *K. pneumoniae* and *S. epidermidis* biofilm matrix, whereas a small amount of peptide was seen in the mid-layers of *P. aeruginosa* biofilm (most being in the surface layer). Probably as a consequence, $IC_{50}$ values against *P. aeruginosa* were higher than other tested bacterial strains. Bacterial species differ in their biofilm matrix components [56]. For instance, the ability of *P. aeruginosa* to synthesize large amounts of the anionic polysaccharide alginate and eDNA probably reduces its susceptibility to magainin II and cecropin P1 and might entrap AMPs (positively charged molecules) before they can reach their bacterial target [57]. Additionally, the positively charged exopolymers (Pel and Psl) in *P. aeruginosa* biofilm can also cause electrostatic repulsion of the cationic polypeptide antibiotics (polymyxin B and colistin) [58], as also observed for human β-defensin-3 (hBD-3) and LL-37 against *S. epidermidis* [59]. Nevertheless, our results still indicate the rapid penetration and strong killing activity of D-LL-31 against all tested isolates in a strain-, biofilm-forming ability-, and matrix component-dependent manner. We can conclude that consideration of the interaction capacity of AMPs with biofilm matrix and interference of biofilm-specific molecules will be an essential part of the development of peptide-based therapeutics.

There is much evidence that AMPs in combination with common antibiotics often enhance the efficacy of the antibiotic and decrease development of antibiotic resistance [60]. In addition, previous reports have shown that biofilm-related infections in CRS patients are more difficult to clear, prone to relapse, inherently resistant to antibiotics and might lead to repeated surgery [32, 34]. In our study, D-LL-31 showed synergistic effects with conventional antibiotics (AMX and TOB) against embedded bacterial cells within biofilm in all tested isolates, leading to greatly reduced $IC_{50}$ values (as much as 16-fold lower) and restoration of the efficacy of the antibiotics. The lowest quantity of biofilm matrix was observed after incubation with a combination of D-LL-31 and antibiotics in all isolates tested. Indeed, D-LL-31 alone could reduce biofilm matrix to a greater extent than did either antibiotic. We hypothesize that the improved antibiotic activity in the presence of D-LL-31 is likely due to the interference with intracellular signals for biofilm formation [61], down-regulation of genes essential for biofilm development [62], reduction of biofilm matrix synthesis [63] and biofilm disruption by the AMPs [15, 64]. Consequently, these may recognize the possible mechanisms that perturbation of matrix synthesis and disruption of biofilm architecture caused by AMPs plays a key role in allowing increased access of currently used antibiotics toward biofilm-embedded bacteria, as also previously expected [15, 64]. These findings demonstrate that enhanced activities of antibiotic in combination with D-LL-31 to eradicate the biofilm matrix and increase the bactericidal efficacy against embedded cells within biofilm may reduce the relapse cases and number of patients for revision endoscopic sinus surgery.

CRS is an inflammatory disorder disease [1, 3]. Given that, we should avoid the use of any treatment that could induce expression of the major cytokines involved in immunopathogenesis of CRS [28]. Thus, the immunomodulatory effect of D-LL-31 was investigated. Previous research has indicated that elevated expression of IL-4, IFN-γ and IL-6 on airway epithelial cells of CRS patients resulted in decreased epithelial barrier function, a phenomenon that might account in part for the progression of CRS [21, 22]. We did not detect IFN-γ and IL-6 in the presence of D-LL-31, while low levels of IL-4 were detected both in the presence and

absence of the peptide. Although the neutrophil-attracting chemokines, IL-8 and GRO-a, b and g, increased following D-LL-31 treatment, this increase was not statistically significant. We also found high levels of ENA-78 in the culture supernatants with D-LL-31. Rudack and colleagues recently noted that IL-8 and ENA-78 appear to be of secondary importance for the chemotaxis of neutrophils in CRS [65]. Another recent study found that LL-37 causes increased IL-6 and IL-8 release from human nasal cells [66]; while, IL-6 (a marker of inflammation in CRSwNP) was not observed after incubation with D-LL-31. In addition, TNF-α has been thought to play a role in the pathogenesis of CRS via the recruitment of B cells [23]. We did not detect TNF-α expression after treatment with D-LL-31. Based on our results, D-LL-31 does not appear to induce the cytokines that play a prominent role in ongoing inflammatory reactions in patients with CRS.

Several studies have shown that a balance between MMPs and their regulators (TIMPs) controls the remodeling and repairing of airway ECM [24]. The elevated expression of MMP-2, -7 and -9, important endopeptidases for degrading the ECM, has been considered to play important roles in the pathogenesis of nasal polyposis in CRS patients [24–26]. Significantly decreased expression of TIMP-1 and -2 has been found in patients with CRS, failing to counterbalance activity of MMPs [25, 27]. This imbalance contributes to the excessive degradation of ECM, formation of pseudocysts, albumin deposition and edema [28]. Interestingly, TIMP -2 was up-regulated in presence of D-LL-31. Perhaps D-LL-31 has a novel role in tissue remodeling, which may prevent the progression and development of nasal polyposis in patients with CRS.

Successful development of any pharmaceutical substance requires minimal or no toxicity. D-LL-31 exhibited no cytotoxicity on human lung epithelial cells after 24 h of incubation. Additionally, D-LL-31 exhibits only marginal hemolytic activity against hRBCs (less than 1%) [15].

## Conclusion

In summary, our study demonstrated that D-LL-31 displayed strong broad-spectrum activity against bacteria in planktonic condition. Furthermore, this peptide exhibited a high potential to kill bacterial cells under biofilm-stimulating conditions when compared to all tested agents. Increased biofilm matrix eradication was observed when D-LL-31 was combined with conventional antibiotics. These findings suggested that D-LL-31 is an interesting candidate peptide for further development as an antibacterial or antibiofilm agent for use alone, or to enhance the efficacy of commonly used antibiotics to combat biofilm-related CRS. Co-treatment with this peptide may permit reduction in antibiotic concentrations, possible side effects and disease severity. D-LL-31 was not toxic to human lung epithelia cells. No cytokines involved in the immunopathogenesis of CRS were induced after treatment with D-LL-31. A role for D-LL-31 in the remodeling and repairing of airway extracellular matrix was also observed. Therefore, D-LL-31 is a feasible therapeutic supplement for applying to the biofilm-associated infection of CRS patients.

## Supporting information

**S1 Table. Antimicrobial susceptibility test results of reference strain and representative bacteria used from CRS patients.** Bacterial suspensions were incubated with antibiotics for 24 h and the results were interpreted according to BD BBL™ Sensi-Disc™ antimicrobial susceptibility test discs.
(DOCX)

## Acknowledgments

We would like to acknowledge Prof. David Blair for editing the MS via Publication Clinic KKU, Thailand.

## Author Contributions

**Conceptualization:** Suwimol Taweechaisupapong, Sakawrat Kanthawong.

**Data curation:** Saharut Wongkaewkhiaw, Sakawrat Kanthawong.

**Formal analysis:** Saharut Wongkaewkhiaw, Sakawrat Kanthawong.

**Funding acquisition:** Saharut Wongkaewkhiaw, Sakawrat Kanthawong.

**Investigation:** Saharut Wongkaewkhiaw, Sanguansak Thanaviratananich, Sakawrat Kanthawong.

**Methodology:** Saharut Wongkaewkhiaw, Suwimol Taweechaisupapong, Jan G. M. Bolscher, Kamran Nazmi, Sakawrat Kanthawong.

**Writing – original draft:** Saharut Wongkaewkhiaw, Sakawrat Kanthawong.

**Writing – review & editing:** Saharut Wongkaewkhiaw, Suwimol Taweechaisupapong, Jan G. M. Bolscher, Chitchanok Anutrakunchai, Sorujsiri Chareonsudjai, Sakawrat Kanthawong.

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
