## [Decision Letter · Decision Letter 0]

14 Sep 2020

PONE-D-20-18804

D-LL-31 enhances biofilm-eradicating effect of currently used antibiotics for chronic rhinosinusitis and its immunomodulatory activity on human lung epithelial cells

PLOS ONE

Dear Dr. KANTHAWONG,

Thank you for submitting your manuscript to PLOS ONE. After careful consideration, we feel that it has merit but does not fully meet PLOS ONE’s publication criteria as it currently stands. Therefore, we invite you to submit a revised version of the manuscript that addresses the points raised during the review process.

Please provide detailed point to point response to the reviewers' comments especially regarding the comments by the reviewer #2.

We look forward to receiving your revised manuscript.

Kind regards,

Y. Peter Di, Ph.D.

Academic Editor

PLOS ONE

Journal Requirements:

Reviewers' comments:

Reviewer's Responses to Questions

**Comments to the Author**

1. Is the manuscript technically sound, and do the data support the conclusions?

Reviewer #1: Yes

Reviewer #2: Yes

2. Has the statistical analysis been performed appropriately and rigorously? 

Reviewer #1: Yes

Reviewer #2: Yes

3. Have the authors made all data underlying the findings in their manuscript fully available?

Reviewer #1: Yes

Reviewer #2: Yes

4. Is the manuscript presented in an intelligible fashion and written in standard English?

Reviewer #1: Yes

Reviewer #2: Yes

5. Review Comments to the Author

Reviewer #1: Dr. Kanthawong continues her thorough characterization of antimicrobial effects of D-LL-31 in clinically relevant biofilm bacterial infections of mucosal surfaces, here chronic rhinosinusitis.

Experiments are carefully conducted and clearly presented. Synergy with antibiotics is demonstrated.

Dr. Kanthawong might consider commenting on anti-viral potential of D-LL-31 and provide references to support this direction for translational investigations. See Yu et al https://doi.org/10.1016/j.isci.2020.100999

Reviewer #2: In this study, Wongkaewkhiaw et al. demonstrated that D-LL-31, the d-enantiomer of LL31 showed significant increase of antimicrobial activities against three pathogens that are associated with chronic rhinosinusitis (CRS). D-LL-31 also showed synergistic effects when combined with clinically used antibiotics. The study design was good in testing the main hypothesis. It was commendable that the authors adopted many relevant methods for the microbiology studies, such as SEM, MIC, MBC, and TSP pegged lids for biofilm studies. Given the urgent need of new antibiotics, I am willing to endorse the good contribution to the field of antimicrobial peptides. However, there are some concerns need to be addressed.

1. There were a lot of experimental details missing from the manuscript and the authors need to provide the required information and clarify the experimental details so the data can be evaluated and reproduced by other researchers. For instance, the AMX or TOB concentrations used in synergistic effect (Table 5) and biofilm images (Figure 4) were not presented.

2. The bacterial strains were isolated directly from patients, which is highly relevant to the study of D-LL-31 within clinical conditions. However, it was not clear how many isolates were used in the subsequent experiments. Did the authors use only one isolate or multiple isolates (from which patients) per bacterial strain for the experiments? Were the antibiogram for the used isolates available and can be provided (even as a supplemental data)? Were the used bacterial isolate(s) representative to the respective bacterial strains? Can the author tried to include some standard laboratory strains such as PAO1 or PA14 for P. aeruginosa so that other researchers in this field can easily compare the activity of D-LL-31 to other AMPs.

3. The biofilm assay description should be clarified regarding the starting bacterial numbers. What were the bacterial numbers used for the biofilm assay? The number seemed to be extremely high for using 200ul of OD600=0.9 bacterial in a 96-well plate well. In addition, the bacterial numbers would also be very different with regards to the three different bacterial strains among PA, KP and SE. Thus, it might not be appropriate to directly compare their biofilm forming ability among different strains as presented in the manuscript.

4. OD readings in MTT assay has been known to not normally being linear to the total cell numbers because of the different growth condition and proliferation rate. A standard curve between OD readings and cell numbers need to be first established before the OD readings can be used for the calculation of cytotoxicity. The authors need to be aware of this inappropriate use of MTT assay.

5. In addition, the cytotoxicity assay for D-LL-31 was also incorrectly performed. The toxicity data should include higher concentration up to 100uM similar to what were used in Fig 3. Furthermore, the toxicity data needs to be factored into the calculation of cytokine inhibition as there was minor cytotoxicity at 10uM, the concentration used in the cytokine study.

6. For figure 1 and table 2, the N of each strains of bacteria should be provided so the data can be interpreted correctly. It is a concern that the sample size may be too small and the results could potentially be biased.

7. Fig 2. As many previous studies have suggested that antimicrobial peptides usually kill bacteria upon contact and the killing is often instant. I wonder why the author picked 1- and 2-hour time points instead of a 1, 5, 10 minutes etc. time frame so that the advantage of peptide instant killing could be showcased. In contrast, AMX and TOB, although effective occasionally, likely wouldn’t work as fast as AMPs such as D-LL-31. The bacterial amount (10^5) used for Figure 2 was less than the standard required numbers, which may affect the correct interpretation of the peptide potency.

8. Why did the MIC values differ from the results presented in Fig 2? If the MIC of AMX against K. pneumoniae is at 4.3uM, why couldn’t it killing the KP in Fig 2 even at 10uM?

9. Fig 3. The AMX and TOB concentrations used were way higher than the clinical MIC breakpoints. Are these concentrations relevant to the CRS treatment?

10. The failed bacterial culture rate was at 50%. Although a discussion regarding the failed bacterial culture could be resulted from other microbial species (anaerobic bacteria and fungi), it could simply be just the culturing techniques/methods. This is because those microbial species (anaerobic bacteria and fungi) should be rare and existed at much lower percentage as discussed and not likely to be accounted for almost 50% of the failed culture. The authors should either provide additional evidence (PCR or sequencing) or to revise the description.

11. For Table 5, what were the applicable concentrations of AMX, TOB, and LL-31 used for determining the synergistic action? Are the AMX concentrations used achievable or relevant in clinical conditions? There are a lot of experimental details missing for an effective evaluation of the presented data. In addition, what was the rationale for testing the synergistic effect if your peptide or TOB was already very effective? Why not choose resistant isolates/strains for evaluating the synergistic effect?

12. Although the binding data of D-LL-31 to the fixed biofilm matrix is interesting, it could not directly explain if the peptide affect the membrane and disrupt the biofilm especially the nature of positive charged D-LL-31 would be expected to bind the negatively charged bacterial membrane regardless if it disrupt the biofilm matrix.

13. Fig 6. A lung cancer cell line A549 was chosen for cell viability studies. A549 is a cancer cell line and it is NOT likely represent the normal nasal epithelial cell response. Can the author justify why A549 is relevant to CRS? Are there any primary nasal epithelial cells available? Also, the epithelial cells were cultured on 96-well plates, which is fine, but there are established air-liquid interface (ALI) cell culture systems for TEER and cell viability studies. The ALI system would be a much better model for the purpose of this study.

14. Fig 7. It looks like the cells were treated with peptides only? Can the author explain why the cells were not first exposed to bacterial infection and then quantify the cytokines? It’s unlikely that a patient will instill the peptide under normal condition for a prophylactic treatment.

15. Line 578 regarding the difference to the LL37. You did not perform the experiment with the available parent peptide LL37 together and this is an overstatement for the results.

Minor concerns:

1. The scale bars of the SEM images were mentioned but not provided, which should be included.

2. The statement “These findings indicated that D-LL-31 was more effective than the L-form and the LL-37 variants against all bacteria grown as biofilm” was incorrect as it only showed better activity for S. epidermidis strain.

3. Line 517, citation or proof showing that D-LL-31 is protease resistant should be provided.

4. A better proofreading should be performed. For instance, line 488, the “enhancement” biofilm-eradicating effect of…..

5. In line 533, what does it mean for “Indeed, D-LL-31 alone could reduce biofilm matrix to a greater extent than did either antibiotic.” It did not seem to be in accordance with the presented data of TOB.

6. PLOS authors have the option to publish the peer review history of their article (what does this mean?). If published, this will include your full peer review and any attached files.

Reviewer #1: No

Reviewer #2: No

---

## [Author Response · Author response to Decision Letter 0]

2 Nov 2020

We would like to thank you for your e-mail dated September 14, 2020, together with the reviewer’s comments. We would like to thank the reviewers for their thoughtful review of the manuscript. They raise important issues and their inputs are very helpful for improving the manuscript. We have carried out the experiments that the reviewers suggested to demonstrate the antimicrobial activity of antibiotic (AMX and TOB) and AMPs (LL-37, LL-31 and their D-form) against P. aeruginosa ATCC27853 (bacterial reference strain) in order to compare the activity of D-LL-31 with other AMPs (Fig 2D), have added S1 Table of the Antimicrobial susceptibility of reference strain and representative bacteria from CRS patients and revised Fig 6 showing the cytotoxicity of D-LL-31 (0.05-100 µM) on human lung epithelial cells. We also have revised our manuscript accordingly. The modified parts of the manuscript are highlighted. Please find attached a point-by-point response to reviewer’s concerns. 

 We consider that the manuscript has improved significantly as a result of the comments. We hope that you find our responses satisfactory and that the manuscript is now acceptable for publication in PLOS ONE.

---

## [Decision Letter · Decision Letter 1]

19 Nov 2020

D-LL-31 enhances biofilm-eradicating effect of currently used antibiotics for chronic rhinosinusitis and its immunomodulatory activity on human lung epithelial cells

PONE-D-20-18804R1

Dear Dr. KANTHAWONG,

We’re pleased to inform you that your manuscript has been judged scientifically suitable for publication and will be formally accepted for publication once it meets all outstanding technical requirements.

Kind regards,

Y. Peter Di, Ph.D.

Academic Editor

PLOS ONE

Additional Editor Comments (optional):

Reviewers' comments:

Reviewer's Responses to Questions

**Comments to the Author**

1. If the authors have adequately addressed your comments raised in a previous round of review and you feel that this manuscript is now acceptable for publication, you may indicate that here to bypass the “Comments to the Author” section, enter your conflict of interest statement in the “Confidential to Editor” section, and submit your "Accept" recommendation.

Reviewer #1: All comments have been addressed

Reviewer #2: All comments have been addressed

2. Is the manuscript technically sound, and do the data support the conclusions?

Reviewer #1: Yes

Reviewer #2: Yes

3. Has the statistical analysis been performed appropriately and rigorously? 

Reviewer #1: Yes

Reviewer #2: Yes

4. Have the authors made all data underlying the findings in their manuscript fully available?

Reviewer #1: Yes

Reviewer #2: Yes

5. Is the manuscript presented in an intelligible fashion and written in standard English?

Reviewer #1: Yes

Reviewer #2: Yes

6. Review Comments to the Author

Reviewer #1: (No Response)

Reviewer #2: (No Response)

7. PLOS authors have the option to publish the peer review history of their article (what does this mean?). If published, this will include your full peer review and any attached files.

Reviewer #1: No

Reviewer #2: No

---

## [Editor Report · Acceptance letter]

26 Nov 2020

PONE-D-20-18804R1 

D-LL-31 enhances biofilm-eradicating effect of currently used antibiotics for chronic rhinosinusitis and its immunomodulatory activity on human lung epithelial cells 

Dear Dr. Kanthawong:

I'm pleased to inform you that your manuscript has been deemed suitable for publication in PLOS ONE. Congratulations! Your manuscript is now with our production department. 

Kind regards, 

on behalf of

Dr. Y. Peter Di 

Academic Editor

PLOS ONE